# Age-Dependent Dysregulation of APP in Neuronal and Skin Cells from Fragile X Individuals

**DOI:** 10.3390/cells12050758

**Published:** 2023-02-27

**Authors:** Giulia Cencelli, Laura Pacini, Anastasia De Luca, Ilenia Messia, Antonietta Gentile, Yunhee Kang, Veronica Nobile, Elisabetta Tabolacci, Peng Jin, Maria Giulia Farace, Claudia Bagni

**Affiliations:** 1Department of Biomedicine and Prevention, Faculty of Medicine, University of Rome Tor Vergata, 00133 Rome, Italy; 2Institute of Neurosurgery, Fondazione Policlinico Universitario A. Gemelli Istituto di Ricovero e Cura a Carattere Scientifico (IRCCS), Catholic University, 00168 Rome, Italy; 3Faculty of Medicine, UniCamillus, Saint Camillus International University of Health and Medical Sciences, 00131 Rome, Italy; 4Istituto di Ricovero e Cura a Carattere Scientifico (IRCCS) San Raffaele Roma, 00166 Rome, Italy; 5Department of Human Genetics, Emory University School of Medicine, Atlanta, GA 30322, USA; 6Institute of Genomic Medicine, Fondazione Policlinico Universitario A. Gemelli Istituto di Ricovero e Cura a Carattere Scientifico (IRCCS), Catholic University, 00168 Rome, Italy; 7Department of Fundamental Neurosciences, Faculty of Biology and Medicine, University of Lausanne, 1005 Lausanne, Switzerland

**Keywords:** Fragile X syndrome, APP processing, protein synthesis, peptide therapy, iPSCs, ADAM10, SAP97

## Abstract

Fragile X syndrome (FXS) is the most common form of monogenic intellectual disability and autism, caused by the absence of the functional fragile X messenger ribonucleoprotein 1 (FMRP). FXS features include increased and dysregulated protein synthesis, observed in both murine and human cells. Altered processing of the amyloid precursor protein (APP), consisting of an excess of soluble APPα (sAPPα), may contribute to this molecular phenotype in mice and human fibroblasts. Here we show an age-dependent dysregulation of APP processing in fibroblasts from FXS individuals, human neural precursor cells derived from induced pluripotent stem cells (iPSCs), and forebrain organoids. Moreover, FXS fibroblasts treated with a cell-permeable peptide that decreases the generation of sAPPα show restored levels of protein synthesis. Our findings suggest the possibility of using cell-based permeable peptides as a future therapeutic approach for FXS during a defined developmental window.

## 1. Introduction

Fragile X syndrome (FXS), an X-linked condition, is the most frequent form of hereditary intellectual disability (ID) and monogenic cause of autism [1]. Individuals with FXS show physical and behavioral features, including intellectual disability, attention-deficit/hyperactivity disorder (ADHD), repetitive behaviors, and anxiety. Reduced social interactions have been reported in FXS individuals diagnosed with autism spectrum disorder (ASD) [2,3,4,5,6,7]. Indeed, around 40% of patients with FXS meet the criteria for ASD [8,9,10].

FXS occurs due to the absence or mutation of fragile X messenger ribonucleoprotein 1 (FMRP). FMRP is an RNA-binding protein involved in several aspects of mRNA metabolism, including the regulation of mRNA translation [11,12,13,14,15]. The absence of FMRP compromises the regulated expression of a variety of proteins critical for brain development, synaptic plasticity, and dendritic spine morphology, ultimately impinging on cognition and behavior [2,12,15]. Consistent with its key role in the brain, FMRP regulates a large subset of mRNAs [12,16,17], including the mRNA encoding amyloid precursor protein (APP) [12,18,19,20,21].

APP is a type I transmembrane protein central to the pathogenesis of Alzheimer’s disease (AD) [22,23]. Besides its well-established role in neurodegeneration, APP also exerts a key role in physiological functions, including synaptogenesis and synaptic plasticity [24,25,26,27,28,29]. APP undergoes a complex series of proteolytic processing events, which can be divided into amyloidogenic and non-amyloidogenic pathways. The amyloidogenic pathway is characterized by the production of Aβ peptides, which are associated with AD progression [23,28]. Under physiological conditions, the non-amyloidogenic pathway results in the release of the soluble amyloid precursor protein-alpha (sAPPα) [24,25,28,30] catalyzed by/driven by the α-secretase activity of the disintegrin and metalloprotease ADAM10 [31,32,33]. The sAPPα fragment, regulates several processes in brain development, including synaptic plasticity, spine density, and cognition [25,30,34,35].

Several points of evidence support the involvement of APP in the FXS phenotype [18,36,37,38,39,40,41]. FMRP mediates mGluR5-dependent translation of *APP* mRNA, and its absence leads to exaggerated APP expression in *Fmr1* KO mice and individuals with FXS [18,20,36,40]. Genetic reduction of APP expression rescues synaptic deficits and behavioral phenotypes in the *Fmr1* KO mice [20,40,41]. In addition, FMRP regulates *Adam10* mRNA translation, and lack of FMRP in mice (*Fmr1* KO) increases both APP and ADAM10 protein levels [18,20]. This dysregulation ultimately leads to excessive production of sAPPα during a specific developmental window in mice (analogous to childhood and early adolescence in human), corresponding to a critical stage of synaptogenesis [20]. Excessive release of sAPPα contributes to some of the main molecular features of FXS, namely increased protein synthesis, aberrant spine morphology, and altered synaptic function and behavior [2,20,42]. Of note, treatment of juvenile *Fmr1* KO mice with a specific cell-permeable peptide (TAT-Pro ADAM10^709−729^) that interferes with ADAM10-mediated APP processing rescues mRNA translation, spine morphology and behavioral defects [20].

Despite the promising results obtained in *Fmr1* KO mice, the use of a rodent model for FXS presents limitations that may hamper the translation of preclinical data to humans. Several clinical trials for FXS, based on findings generated in the mouse model, have not been very successful so far [4,5,43,44], suggesting that models using human patient-derived cells will be important for the development of new and personalized therapies.

In the present study, we analyzed the processing of APP in FXS human fibroblasts, neurons derived from human induced pluripotent stem cells (iPSCs), and human forebrain organoids. We observed a specific age-dependent dysregulation of APP metabolism in human cells. In addition, treatment of FXS fibroblasts with the cell-permeable TAT-Pro ADAM10^709−729^ peptide reduces sAPPα release and normalizes the level of protein synthesis. These findings suggest that a subset of individuals with FXS, those with elevated protein synthesis, could benefit from a peptide therapy based on the reduction of excessive sAPPα.

## 2. Materials and Methods

### 2.1. Human Fibroblasts and iPSCs Cultures

Fibroblasts. Control fibroblast cell lines (*n* = 19, age range 5–57 years) were purchased from the Coriell Cell Repositories. FXS fibroblast cell lines (*n* = 32, age range 6–69 years) were obtained from dermal biopsies with patient consent and under approval from multiple centers as listed in Appendix A (CHUV University Hospital of Lausanne; M.I.N.D. Institute in Sacramento; Erasmus Medical Center in Rotterdam and University Hospital A. Gemelli in Rome). The clinical assessment, inclusion criteria, study protocol, the *FMR1* mRNA and FMRP levels, and all amendments for Switzerland, USA, and Netherlands cohorts have been previously described [42]. The samples collected at the University Hospital A. Gemelli in Rome were derived from 4 FXS individuals (ET001, ET002, ET003, and ET004). CGG sizing and methylation status were evaluated using AmplideX^®^ PCR and AmplideX^®^ mPCR assays (Asuragen, Austin, TX, USA) or by Southern blot analysis using *Hind*III restriction enzyme and/or the methylation-sensitive enzyme *Eag*I (New England Biolabs, Ipswich, MA, USA). The study protocol was approved by the Ethics Committee of the University Hospital A. Gemelli in Rome (prot. N. 9917/15 and prot.cm 10/15). The level of *FMR1* mRNA and FMRP for these lines are included in Appendix A.

Fibroblasts were maintained in DMEM/F-12 (Gibco, Thermo Fisher Scientific, Waltham, MA, USA) supplemented with 10% fetal bovine serum (Gibco, Thermo Fisher Scientific), 1X GlutaMax^TM^ (Gibco, Thermo Fisher Scientific), 1X penicillin-streptomycin (Gibco, Thermo Fisher Scientific) and MycoZap reagent (Lonza, Basel, Switzerland).

Induced pluripotent stem cells (iPSCs). iPSCs derived from fibroblasts of typically developing individuals (TDI) and FXS individuals were established at the Children’s Hospital of Orange County and kindly provided by Dr. Philip H. Schwartz [45]. iPSCs were cultured on Matrigel (BD Biosciences, Franklin Lakes, NJ, USA) in mTeSR medium (Stem Cell Technologies, Vancouver, BC, Canada). The clinical characteristics of iPSCs used in the present study are summarized in Appendix A. The relative levels of *FMR1* mRNA and FMRP are shown in Appendix A.

Forebrain organoids. Human forebrain organoids used in this study were generated from TDI and FXS iPSCs at Emory University School of Medicine in Atlanta and previously characterized [46]. iPSCs were cultured on irradiated mouse embryonic fibroblasts (MEFs) in human iPSC medium consisting of DMEM/F-12 (Gibco, Thermo Fisher Scientific), 20% knockout serum replacement (KSR, Gibco, Thermo Fisher Scientific), 1X GlutaMAX (Gibco, Thermo Fisher Scientific), 1X MEM non-essential amino acids (Gibco, Thermo Fisher Scientific), 100 μM β-mercaptoethanol (Gibco, Thermo Fisher Scientific), and 10 ng/mL human basic FGF (PeproTech, London, UK).

### 2.2. iPSC Neural Differentiation

iPSCs were differentiated into neurons as previously described [47]. Briefly, iPSCs were maintained in culture in defined default media (DDM) consisting of DMEM/F-12 supplemented with 1X N-2 supplement (Gibco, Thermo Fisher Scientific), 1X B-27 supplement (Gibco, Thermo Fisher Scientific), bovine albumin fraction V 7.5% (Gibco, Thermo Fisher Scientific), 1X MEM non-essential amino acids (Gibco, Thermo Fisher Scientific), 1 mM sodium pyruvate (Gibco, Thermo Fisher Scientific), 100 μM β-mercaptoethanol (Gibco, Thermo Fisher Scientific), and 100 ng/mL human recombinant Noggin (Stem Cell Technologies) with a daily medium change [47,48]. After 16 days, the medium was changed to DDM, supplemented with B-27 supplement (Gibco, Thermo Fisher Scientific) without recombinant Noggin. After 24 days, cells were dissociated and plated into poly-ornithine/laminin-coated wells. Five to seven days after dissociation, half of the medium was replaced with neurobasal (Gibco, Thermo Fisher Scientific) supplemented with 1X B-27 supplement (Gibco, Thermo Fisher Scientific) and 2 mM glutamine (Gibco, Thermo Fisher Scientific).

### 2.3. Human Forebrain-Specific Organoid Cultures

Forebrain-specific organoids were generated using established protocols as previously described [46,49]. Human iPSC colonies were detached from the MEF feeder layer with 1 mg/mL collagenase treatment for 1 h and suspended in embryonic body (EB) medium, consisting of FGF-2-free iPSC medium supplemented with 2 μM dorsomorphin (MilliporeSigma, Burlington, MA, USA) and 2 μM A-83 (Tocris Bioscience, Bristol, UK) in non-treated polystyrene plates for 4 days with a daily medium change. On days 5–6, half of the medium was replaced with induction medium consisting of DMEM/F-12, 1X N-2 supplement (Gibco, Thermo Fisher Scientific), 10 μg/mL heparin (MilliporeSigma) 1X penicillin/streptomycin, 1X MEM non-essential amino acids (Gibco, Thermo Fisher Scientific), 1X GlutaMAX (Gibco, Thermo Fisher Scientific), 4 ng/mL WNT-3A (R&D Systems, Minneapolis, MN, USA), 1 μM CHIR99021 (Tocris Bioscience), and 1 μM SB-431542 (Tocris Bioscience). On day 7, organoids were embedded in Matrigel (BD Biosciences) and grown in the induction medium for 6 more days. On day 14, embedded organoids were mechanically dissociated from Matrigel by pipetting onto the plate with a 5 mL pipette tip. Typically, 10–20 organoids were transferred to each well of a 12-well spinning bioreactor (SpinΩ) containing differentiation medium, consisting of DMEM/F-12, 1X N-2, and B-27 supplements (Gibco, Thermo Fisher Scientific), 1X penicillin/streptomycin, 100 μM β-mercaptoethanol (Gibco, Thermo Fisher Scientific), 1X MEM non-essential amino acids (Gibco, Thermo Fisher Scientific), and 2.5 μg/mL insulin (MilliporeSigma). Media was changed every other day.

### 2.4. SUnSET Assay

Protein synthesis assays were performed as previously described using the surface sensing of translation (SUnSET) technique [42,50]. Briefly, cells (80,000/well) were seeded on 12-multiwell plate wells and incubated with 5 µg/mL puromycin (Merck, Darmstadt, Germany) for 30 min, chased with fresh complete medium for 15 min, and then lysed. Cell lysates were analyzed for puromycin incorporation by Western blotting using a specific antibody against puromycin (PMY-2A4, DSHB, Iowa City, IA, USA). Coomassie staining of total proteins was used as a loading control.

### 2.5. Ammonium Sulfate Precipitation

For protein precipitation, 1.5 volumes of saturated ammonium sulfate (according to [51]) was added to the cell media for protein extraction. Proteins were precipitated by centrifugation at max speed for 20 min, and the pellet was resuspended in Laemmli buffer.

### 2.6. Peptide Treatment

TAT-Pro ADAM10^709−729^ and TAT-Ala ADAM10^709−729^ peptides were produced by Peptide 2.0 Inc. (https://www.peptide2.com/) and resuspended in sterile H_2_O. Cells were treated with 20 µM TAT-Pro or TAT-Ala peptide, added to the medium. After 18 h, protein extracts were prepared from the collected cell medium.

### 2.7. Western Blot

Standard methodologies were used. Protein extracts were separated by 10% or 8% SDS-PAGE and transferred to a PVDF membrane. Membranes were incubated using the following specific antibodies, including mouse anti-puromycin (1:500, DSHB), mouse anti-Vinculin (1:2000, Merck), mouse anti-GAPDH (1:2000, Invitrogen, Thermo Fisher Scientific, Waltham, MA, USA), rabbit anti-APP (1:2000, Merck), rabbit anti-ADAM10 (1:500, Abcam, Cambridge, UK), mouse anti-sAPPα (1:500, IBL America, Minneapolis, MN, USA), rabbit anti-OCT3/4 (1:1000, Santa Cruz Biotechnology, Dallas, TX, USA), mouse anti-MAP2 (1:2000, Merck), mouse anti-Nestin (1:1000 Santa Cruz Biotechnology), mouse anti-SAP97 (1:1000, ENZO Life Sciences, Farmingdale, NY, USA) and rabbit anti-FMRP (1:1000, produced in house PZ1 [52]), HRP-conjugated anti-rabbit and anti-mouse secondary antibodies (1:5000, Cell Signaling Technology, Danvers, MA, USA). Proteins were revealed using an enhanced chemiluminescence kit (Bio-Rad, Hercules, CA, USA) and the imaging system LAS-4000 mini (GE Healthcare, Chicago, IL, USA). Quantification was performed using the IQ ImageQuant TL software (GE Healthcare). Detection of GAPDH, Vinculin, and Coomassie staining were used as normalizers. For all SDS-PAGE PageRuler™ Plus Prestained Protein Ladder (10 to 250 kDa, Thermo Fisher Scientific) was used.

### 2.8. RT-qPCR

Total RNA was extracted with TRIzol according to the manufacturer’s protocol (Invitrogen, Thermo Fischer Scientific). For the synthesis of cDNA, 500 ng of total RNA was used. mRNAs were quantified by real-time PCR using SYBR^®^ Green Master Mix (Bio-Rad) on StepOnePlus™ Real-Time PCR machine (Applied Biosystems, Thermo Fischer Scientific, Waltham, MA, USA) according to the manufacturer’s instructions using specific primers. mRNA levels were expressed as relative abundance compared to *HPRT1* and *GAPDH* mRNAs using the (2^−ΔΔCT^) method. The primers used for the amplification of the selected human genes are:*hFMR1 Forward*  *5′-TGT CAG ATT CCC ACC TCC TG-3′**hFMR1 Reverse*  *5′-TAA CCA CCA ACA GCA AGG CT-3′**hHPRT1 Forward*   *5′-TGC TGA GGA TTT GGA AAG GGT-3′**hHPRT1 Reverse*  *5′-TCG AGC AAG ACG TTC AGT CC-3′**hGAPDH Forward*  *5′-CTC AAC TAC ATG GTT TAC ATG-3′**hGAPDH Reverse*  *5′-CCA TTG ATG ACA AGC TTC CCG-3′*

### 2.9. Statistics

Sample size calculation was performed based on the level of sAPPα in fibroblasts measured in a preliminary study. We determined the need for a sample size of at least *n* = 16/TDI and *n* = 28/FXS with a power of 80%, alpha = 0.05, and effect size d = 0.92. The analysis was performed using G*Power 3 [53]. Statistical analysis was performed with Prism GraphPad version 5.0. Data distribution was tested for normality using the Kolmogorov–Smirnov test. Non-normally distributed data were analyzed through non-parametric tests. The significance level was established at *p* < 0.05. Differences between the two groups were analyzed using an unpaired Mann–Whitney test. The correlation was assessed by Spearman’s correlation test. Two-way ANOVA without repeated measures, followed by Sidak’s multiple comparisons test, was performed to examine the effect of genotype and treatment and their interaction. All data are expressed as mean ± SEM and as fold change relative to TDI.

## 3. Results

### 3.1. Age-Dependent Dysregulation of APP Processing in Human FXS Fibroblasts

Previous work in mice revealed that impaired processing of APP leads to excessive production of sAPPα, at a critical developmental period (Post-natal day 21, P21), contributing to molecular, cellular, and behavioral FXS phenotypes [18,20,40]. An overall increase in APP, ADAM10, and sAPPα was previously reported in a small sample of FXS fibroblasts compared to controls [20].

Here we addressed the contribution of age and cell type specificity to the dysregulation of APP processing in human cells derived from FXS individuals. Specifically, we analyzed APP metabolism in cells derived from a large cohort of FXS subjects with different ages (*n* = 32; age range 6–69 years) and typically developing individuals (TDI *n* = 19; age range 5–57 years) (Figure 1). FXS individuals and TDI were subdivided in three different groups: group 1 with an age below 20 years (TDI *n* = 10; FXS *n* = 10); group 2 between 20 and 30 years (TDI *n* = 5; FXS *n* = 11); group 3 above 30 years (TDI *n* = 4; FXS *n* = 11). The expression of APP, the α-secretase ADAM10 and, as control, FMRP was analyzed in cellular extracts, while the release of sAPPα was assessed in cell media. Increased levels of sAPPα and ADAM10 were observed only in FXS individuals belonging to groups 1 and 2 (<30 years) (Figure 1A,B) compared to age-matched controls, whereas no significant differences were detected in group 3 (>30 years) (Figure 1C). The level of full-length APP remained elevated in all FXS conditions, independently of age as in [20]. In conclusion, the dysregulation of APP processing is age-dependent and in FXS primary somatic cells appears during the first three decades of postnatal life.

Next, we investigated whether the dysregulation of APP processing was associated with disease severity. Based on Vineland adaptive behavioral scale (VABS) scores available for 18 FXS individuals [42] (see details in Material and Methods and Appendix A) we analyzed correlations between the levels of sAPPα and the scores in four VABS main domains, namely daily living, communication, adaptive behavior, and socialization. Data on motor skills were available only for a few individuals and were not included in the analysis. We found that sAPPα levels negatively correlated with daily living score for the entire cohort, while no significant correlation was detected with other VABS domains (Figure 2).

### 3.2. Peptide-Based Cellular Therapy Reduces sAPPα Release and Exaggerated Protein Synthesis

Previous work showed that the TAT-Pro ADAM10^709–729^ peptide (TAT-Pro) blocks the interaction of ADAM10 with the synapse-associated protein 97 (SAP97), thereby reducing ADAM10 localization at the cell surface [54,55,56] (Figure 3A). In addition, we have previously shown that the modulation of ADAM10 activity and APP processing with TAT-Pro peptide restored excessive protein synthesis and rescued key behavioral deficits in *Fmr1* KO mice [20].

Here we found that SAP97 expression in human fibroblasts is significantly increased in FXS fibroblasts compared to TDI (TDI *n* = 8; FXS *n* = 8: Figure 3B), further supporting the relevance of the SAP97-ADAM10 interaction in FXS neuronal and non-neuronal cells. Next, we investigated the effects of the TAT-Pro peptide in FXS fibroblasts. To define optimal conditions for TAT-Pro peptide treatment, a FXS fibroblast cell line secreting a high level of sAPPα (ID: 94E0363 described in Appendix A and [42]) was treated with different concentrations of peptide (5, 10, and 20 µM) and the amount of sAPPα released in the cell media was measured by Western blot at different time points, i.e., 6, 12 and 18 h. Optimal reduction of released sAPPα was obtained after a treatment with 20 μM TAT-Pro peptide for 18 h (Appendix A).

Fibroblasts derived from 10 FXS subjects and 6 TDI were treated with control (TAT-Ala) or specific (TAT-Pro) peptides. TAT-Pro peptide treatment caused an significant reduction of sAPPα release in FXS fibroblasts, while no differences were observed in the control group (Figure 3C), suggesting a specific effect on FXS cells with altered APP processing.

Since sAPPα levels affect brain protein synthesis [20,57,58,59,60,61], we analyzed the level of mRNA translation in fibroblasts upon treatment with TAT-Pro peptide. Considering that not all FXS individuals exhibit increased protein synthesis and that such variability does not appear to be age-dependent [42,62], we stratified FXS fibroblasts according to their rate of protein synthesis. Fibroblast lines showing 50% more puromycin incorporation than the average in TDI cell lines were classified as “high protein synthesis”. Remarkably, the treatment of FXS cells with TAT-Pro peptide decreased protein synthesis to levels comparable to control cells—specifically in the subset of patients with a higher translation rate (*n* = 5) (Figure 3D). No significant effects were observed in the subgroup of FXS individuals with levels of mRNA translation comparable to controls (*n* = 5) or in the control (TDI) group itself (*n* = 6) (Figure 3D).

Overall, these findings show the specificity of action of TAT-Pro peptide on a well-defined subgroup of FXS individuals with possible implications for therapy.

### 3.3. Excess of sAPPα Is Detected in Differentiating FXS iPSCs and Forebrain Organoids

Fibroblasts are well suited to age-dependent studies, since they retain the epigenetic imprinting of gene expression based on donor’s age [63,64,65]. However, addressing the role of sAPPα in patient-derived cellular models, such as neurons differentiated from iPSCs and forebrain organoids, represent a necessary step forward to validate the relevance of this pathway in FXS.

Cortical neurons derived from FXS and control iPSCs were obtained using a well-established protocol [47]. Neuronal differentiation was monitored following the cellular morphology and evaluating the expression of specific pluripotency and neuronal markers (Figure 4A,B). During neural differentiation (day 0 iPSCs, day 6, day 24 neural precursor cells (NPCs) and day 60 neurons), a gradual reduction of the pluripotency marker OCT3/4 and a progressive appearance of cortical neural progenitor (Nestin) and neuronal (MAP2) markers were observed (Figure 4B). The release of sAPPα was analyzed in vitro on different days after neuronal differentiation in cells derived from 3 FXS and 3 TDI. Protein levels were measured at specific stages: iPSCs (day 0), neural precursors cells (NPCs) (day 19 and day 24), and neurons (day 60) (Figure 4C–F). A significant increase in sAPPα release was observed in the media of FXS NPCs at day 24 compared to control media, while no significant genotype-dependent difference was detected in iPSCs, NPCs at day 19, or mature neurons (Figure 4C–F).

Finally, we evaluated APP processing in human forebrain organoids derived from TDI and FXS. APP and sAPPα expression were analyzed at two different developmental stages—day 30 and day 69—in both TDI and FXS human forebrain organoids. While APP expression was increased in FXS organoids regardless of the developmental stage, the levels of sAPPα were specifically upregulated in the early phase of forebrain development (Appendix A), consistent with the results obtained in the 2D stem cell model (Figure 4).

## 4. Discussion

Several FXS clinical and behavioral phenotypes, such as attention and social deficits, aggressive behavior, and brain structural abnormalities, undergo considerable changes during development [7,66,67,68,69,70,71,72,73]. Therefore, a better understanding of the time window during which the dysregulation of specific molecular pathways occurs might help to design more precise therapeutic interventions.

Here, we demonstrated that the dysregulation of APP processing occurs in an age-dependent manner in three different human FXS cellular models and that such dysregulation can be targeted by using a specific cell-permeable peptide. Particularly, we observed increased levels of released sAPPα in fibroblasts derived from young FXS individuals, iPSC-derived NPCs, and early-stage forebrain organoids.

*Age-dependent dysregulation of APP processing in FXS fibroblasts and NPCs*. sAPPα plays a key role in processes that are crucial for proper brain structure and function, such as synaptogenesis, synaptic plasticity, protein synthesis, and ultimately, memory formation [28,34,35,60,61,74,75]. The dysregulated release of sAPPα may affect critical neuronal functions and, ultimately, lead to neurodevelopmental defects. Consistent with this proposed mechanism, increased sAPPα levels have been found in the plasma of pediatric FXS subjects [38] and in juvenile *Fmr1* KO mice [20]. In contrast, increased Aβ levels have been observed in plasma and post-mortem brain samples of adult FXS individuals and in adult *Fmr1* KO mice [18,20,37,76], supporting the development-dependent dysregulation of APP processing in FXS. Of note, high levels of sAPPα have also been reported in the plasma of juvenile idiopathic autistic patients [37,38,39,77,78,79,80,81], in ASD individuals with severe clinical manifestations [78,79,80] and subjects with Angelman syndrome [82], suggesting that sAPPα may play a critical role in the pathogenesis of different neurodevelopmental disorders.

The reprogramming and differentiation of patient-derived cells into neurons allows investigators to model/duplicate some of the cellular and molecular features of neurodevelopmental disorders [83,84,85]. Neurons and brain organoids derived from FXS iPSCs recapitulate several cellular pathological aspects reported in the murine model, such as deficits in neurite initiation and outgrowth, increased protein synthesis, neuronal hyperactivity, and deficits in action potential firing and spontaneous synaptic activity [46,86,87,88,89,90,91,92,93,94,95,96]. Moreover, FXS NPCs showed increased proliferation and protein synthesis [46,86,89,96], and several studies demonstrated the involvement of sAPPα in the proliferation and differentiation of murine NPCs [97,98,99,100]. In addition, a study performed on iPSCs derived from TDI reported that non-amyloidogenic processing of APP occurs predominantly at the early stages of neurogenesis [101]. sAPPα, therefore, plays an important role in the initial step of neuronal induction for proper brain development. Here, we observed an excess of sAPPα release in FXS NPCs derived by iPSCs in both 2D and 3D cellular models, which may underlie some of the pathological FXS phenotypes observed in NPCs [46,86,96] and the long-lasting defects reported in the mature neurons [90].

*Contribution of sAPPα to FXS clinical manifestations.* Although the number of individuals with FXS included in our analysis is limited, our findings showing a negative correlation between the levels of sAPPα and Vineland scale scores, -- specifically, the daily living score --suggest that the levels of sAPPα may be predictive of the clinical outcome. We based the present study on a heterogeneous group of FXS individuals from four cohorts from different countries (see Appendix A) and used standardized methods for the molecular analyses. Additional multicentric studies that include a longitudinal follow-up will be valuable and necessary to further validate the presence of excessive sAPPα during a defined developmental window to consider sAPPα as a biomarker for FXS.

Some FXS clinical features show changes in severity across ages; deficits in social behavior, for example, appear to improve during development [7,70,71,72]. It is tempting to hypothesize that age-dependent dysregulation of sAPPα could contribute to this clinical manifestation. In agreement with this, data generated in the mouse model of FXS showed that the age-dependent upregulation of sAPPα has an effect on a measure/aspect of social activity (nest building) that is rescued by decreasing the excess of sAPPα [20]. In addition, a clinical manifestation that tends to resolve over time is the presence of seizures [6,102]. Interestingly, seizures have also been reported in individuals with ASD [103,104] and with Angelman syndrome [105,106], which share with FXS the excessive sAPPα production [78,79,80,82].

*TAT-Pro peptide treatment as a therapeutic strategy for FXS.* Despite the efforts to understand the pathophysiology of FXS, to date, there is still no effective treatment for this disorder. Reduction of sAPPα levels by modulating its receptor activity and downstream-activated pathways might represent a valid approach to reducing the pathological effects of exaggerated sAPPα release. Nevertheless, no specific receptor for sAPPα has been currently identified. The postsynaptic α7 nicotinic acetylcholine receptor (nAChR) [107] and GABA_B_R1_a_ have been recently proposed as candidates for sAPPα receptors [108,109]. However, both receptors have different crucial functions in the brain and are not specific for sAPPα, therefore their modulation may have deleterious consequences in FXS [110].

The use of cell-permeable peptides represents a new promising therapeutic strategy for precise medicine, and several peptides are currently being tested in clinical trials [111,112,113,114,115]. ADAM10 activity can be modulated using a specific cell-permeable peptide, which is able to reduce ADAM10 localization to the membrane [55]. We previously demonstrated that modulation of ADAM10 activity using TAT-Pro peptide normalizes sAPPα levels in *Fmr1* KO mice and ameliorates various molecular, synaptic, and behavioral defects, including exaggerated protein synthesis, enhanced mGluR-dependent long-term depression (LTD), maternal/social skills, memory, and hyperactivity [20]. In addition, the same TAT-Pro peptide treatment has been demonstrated effective in rescuing cognitive decline in a murine model for Huntington’s disease [56]. 

Here, we tested and demonstrated the validity of a peptide-based strategy in human FXS fibroblasts. Although the iPSC-derived NPCs and neurons represent a suitable tool for drug discovery [116,117], the reprogramming of adult somatic cells to the stem cell state seems to be independent of the age of the individuals [118]. In contrast, fibroblasts retain the epigenetic memory of the donor’s age [63,64,65], allowing the identification of subgroups of FXS that may benefit from the use of the TAT-Pro peptide.

Aberrant mRNA translation represents one of the major hallmarks of FXS and, therefore, a putative therapeutic target [2,14,119]. While therapies aimed at rescuing protein synthesis have provided successful results in mice [120], similar approaches have failed in clinical trials, highlighting the difficulties of translating data obtained in murine models to the clinic [4,5,43,44,66,121]. Several factors may explain these failures, including the lack of patient stratification, varying ages of the enrolled FXS subjects, and the validity of the outcome measures [4,43,66]. In this context, we reported that the dysregulation of protein synthesis is observed only in a subset of patient-derived fibroblasts [42]. We observed a positive effect of the TAT-Pro peptide treatment on protein translation, specifically in FXS cells derived from children/adolescents with a high translation rate. Our findings supporting the hypothesis that targeting protein synthesis, based on patient’s stratification, may be a valid outcome measure in future clinical trials based on personalized medicine [4]. Nevertheless, the restoration of sAPPα levels upon peptide treatment should be carefully monitored over time to maintain sufficient sAPPα levels. Indeed, an excessive reduction of ADAM10 activity has been linked to learning deficits, altered spine morphology, defective synaptic functions, and increased formation of Aβ peptides in mice [122]. Furthermore, subchronic treatment of WT mice with the TAT-Pro peptide has been used to generate a model of sporadic AD that mimics the events occurring in the disease, including β-amyloid aggregate production [54]. Although TAT-Pro peptide is not currently used in clinical trials, our results in human cells, as well as its ability to cross the blood-brain barrier in vivo [20,56], support the possible application of TAT-Pro peptide as a new therapeutic approach for a subgroup of individuals with FXS.

## 5. Conclusions

Overall, our study demonstrates an age-dependent regulation of APP metabolism in different FXS cellular models (fibroblasts, iPSCs, and brain organoids). Particularly, sAPPα is involved in the APP-mediated increase in protein synthesis in FXS, supporting the critical role of APP processing in the pathophysiology of FXS. This work identifies the early stages of childhood and adolescence in humans as the crucial time window for therapeutic intervention based on the restoration of protein homeostasis following the regulation of sAPPα release, with possible long-lasting effects. Our findings suggest that APP may therefore represent a new therapeutic target and/or biomarker for FXS and for other neurodevelopmental disorders and intellectual disabilities, such as ASD, that share with FXS the dysregulation of APP processing.

## Figures and Tables

**Figure 1 cells-12-00758-f001:**
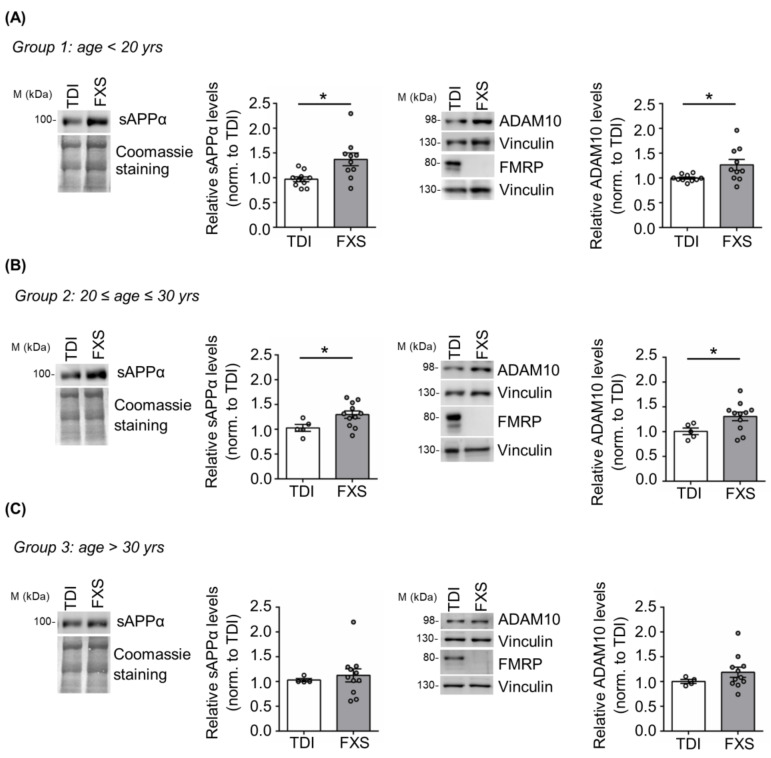
APP processing in human TDI and FXS fibroblasts. Representative Western blot showing sAPPα, ADAM10, and FMRP in TDI and FXS individuals stratified according to the age: (**A**) <20 years (TDI *n* = 10; FXS *n* = 10), (**B**) between 20 and 30 years (TDI *n* = 5; FXS *n* = 11) and (**C**) >30 years old (TDI *n* = 4; FXS *n* = 11). The bar plots show the quantification of sAPPα and ADAM10 levels in TDI and FXS fibroblast cell lines normalized to Coomassie staining and Vinculin, respectively. Error bars represent the SEM (* *p* < 0.05, Mann–Whitney test). ADAM10 and FMRP proteins were detected on separate blots, each with its own control Vinculin.

**Figure 2 cells-12-00758-f002:**
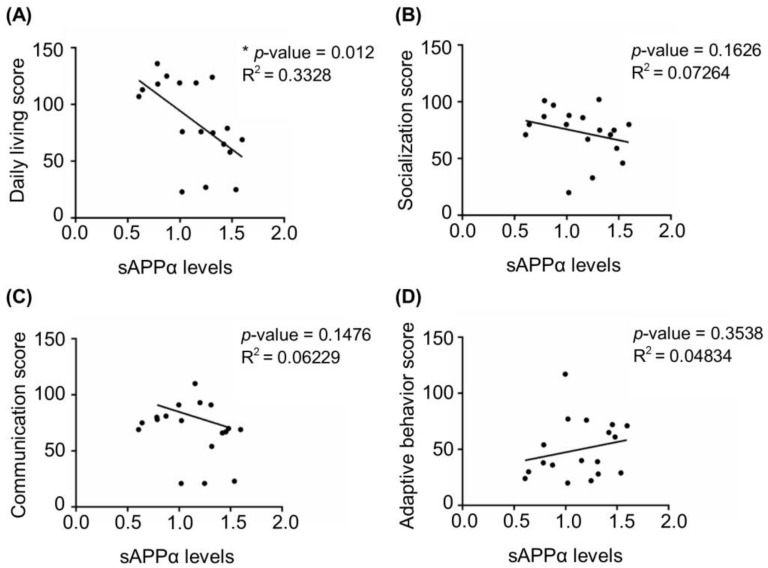
Correlation between four main domains of Vineland adaptive behavior scale and sAPPα levels in FXS. (**A**) Daily living score, (**B**) socialization score, (**C**) communication score, and (**D**) adaptive behavior score of FXS individuals were plotted against the levels of sAPPα detected in fibroblasts from 18 individuals described in Figure 1 (see Materials and Methods and Appendix A for details)(* *p* < 0.05, Spearman correlation coefficient).

**Figure 3 cells-12-00758-f003:**
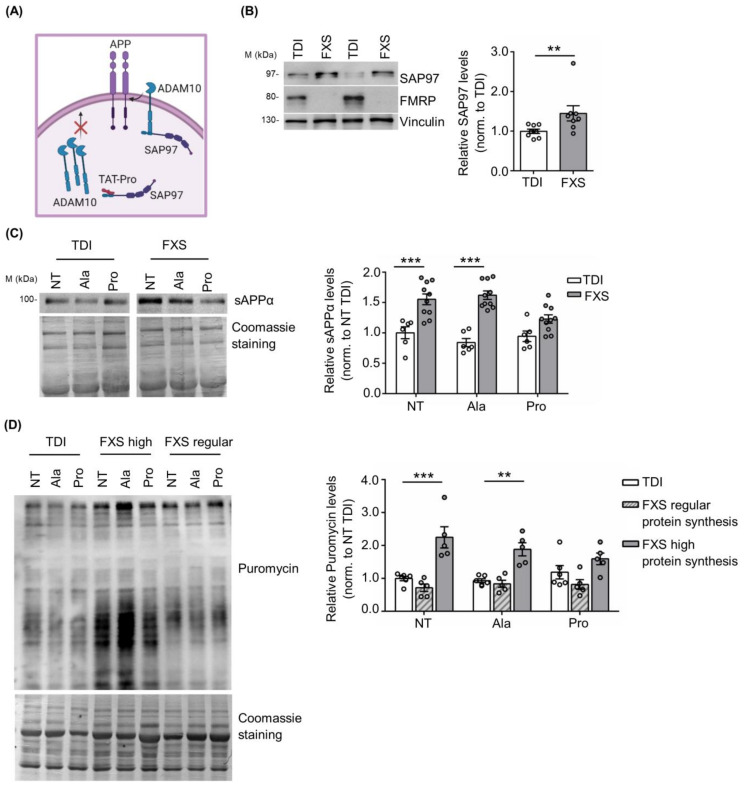
Modulation of ADAM10 in human TDI and FXS fibroblasts. (**A**) Schematic representation of the mechanism of action of the TAT-Pro peptide (image created in BioRender.com). (**B**) Left, representative Western blot showing SAP97 expression in TDI and FXS fibroblasts. Right, the bar plot shows the quantification of SAP97 levels in TDI and FXS fibroblasts (TDI *n* = 8; FXS *n* = 8). Error bars represent the SEM (** *p* < 0.01, Mann–Whitney test). (**C**) Left, representative Western blots showing the levels of sAPPα in non-treated fibroblasts (NT), treated with the control peptide (Ala) or with the specific peptide (Pro). Right, the bar plots show the quantification of sAPPα normalized to Coomassie staining (TDI *n* = 6, FXS *n* = 10). Error bars represent the SEM (Two-way ANOVA analysis: genotype effect F (1, 42) = 61.97 *** *p* < 0.001; treatment effect F(2, 42) = 2.866 *p* > 0.05; interaction effect F(2, 42) = 4.422) (*** *p* < 0.001 NT TDI vs. NT FXS; *** *p* < 0.001 TAT-Ala TDI vs. TAT-Ala FXS, Sidak’s multiple comparisons test). (**D**) Left, representative Western blot showing the levels of puromycin incorporation in fibroblasts NT, treated with the Ala or Pro peptides. Right, the bar plots show the quantification of puromycin normalized to Coomassie staining in TDI and FXS fibroblasts expressing regular or high levels of protein synthesis (TDI *n* = 6, FXS high *n* = 5; FXS regular *n* = 5). Error bars represent the SEM (Two-way ANOVA analysis: genotype effect F(2, 39) = 34.87 *** *p* < 0.001; treatment effect F(2, 39) = 0.4527 *p* > 0.05; interaction effect F(4, 39) = 2.008 *p* > 0.05) (*** *p* < 0.001 NT TDI vs. NT FXS; ** *p* < 0.01 TAT-Ala TDI vs. TAT-Ala FXS, Sidak’s multiple comparisons test).

**Figure 4 cells-12-00758-f004:**
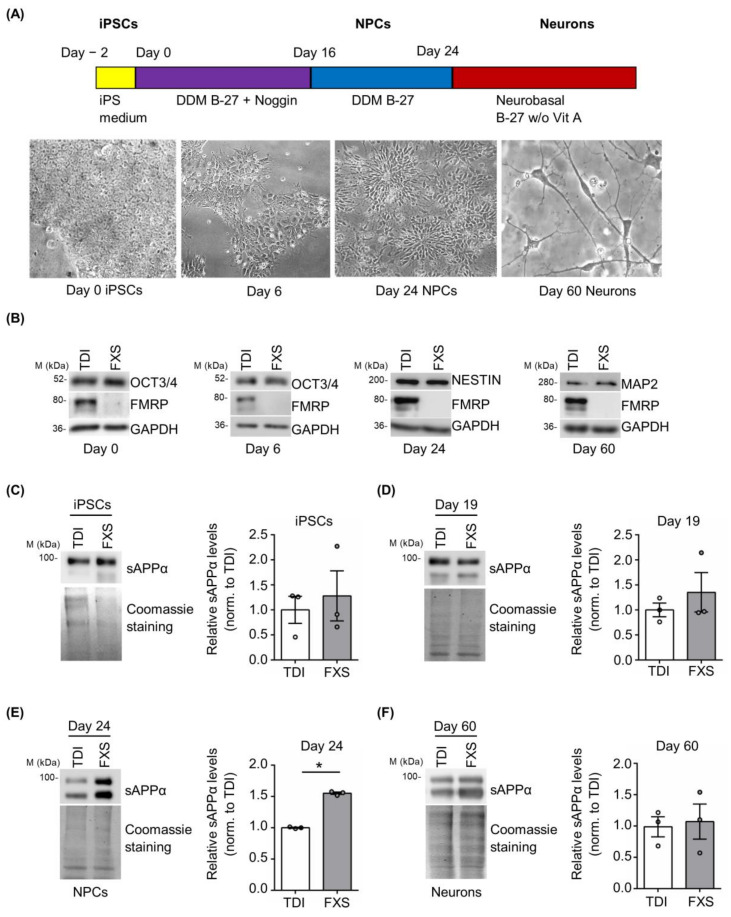
Expression of sAPPα during neuronal differentiation. (**A**) Schematic representation of the neural differentiation and representative bright-field microscopy images of differentiating cells. (**B**) Representative Western blots showing the expression of differentiation markers (OCT3/4, Nestin, and MAP2) during neurogenesis at day 0, day 6, day 24, and day 60. (**C**–**F**) Left, representative Western blots showing the level of sAPPα release in the cell media. Protein levels were measured at several neuronal differentiation stages: (**C**) iPSCs, (**D**) NPCs day 19, (**E**) NPCs day 24, and (**F**) neurons day 60 (TDI *n* = 3; FXS *n* = 3). Right, the bar plots show protein quantification normalized to Coomassie staining. Error bars represent the SEM (* *p* < 0.05 Mann–Whitney test).

## Data Availability

All data supporting the findings of this study are available within the article and its Appendix A file or from the corresponding author upon request.

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
