# Peer review of "Age-Dependent Dysregulation of APP in Neuronal and Skin Cells from Fragile X Individuals"

_cells, 2023, doi:10.3390/cells12050758_

Round 1
Reviewer 1 Report
The researcher demonstrated an age-dependent regulation of APP metabolism in different FXS cellular models (fibroblasts, iPSCs, and brain organoids). Particularly, sAPPα is involved in the APP-mediated increase of protein synthesis in FXS. This work identified the early stages of childhood and adolescence in humans as crucial time window for therapeutic intervention, based on the restoring of protein homeostasis acting on the regulation of sAPPα release. More importantly, they proved that a subset of FXS individuals could benefit from treatment with the cell-permeable peptide TAT-Pro ADAM10709-729 able to specifically reduce sAPPα release normalizing protein synthesis.
The experiments conducted were appropriate for their hypothesis and the figures illustrate very clean results, which are very consistent across the cellular models, not only in the qualitative areas but also related to the timing/developing of the tissues.
If there were any negative results in other experiments, this should be briefly included. Please include a reason for not including morphology and electrophysiology data, given that this pathway was associated with hyperexcitability.
The statistical analysis is lacking sample size calculations and the 2.9 Statistics section should be expanded to include p-value corrections and other details about the analysis.
The authors stated clearly the limitations of the study, I will appreciate a statement about the rigor and reproducibility of the research. There may be a need for another lab to replicate these results and this should be addressed/mentioned in the paper.
While the authors try to link the levels of sAPPα with severity, it is hard to make a biological link between the age-related sAPPα defect and the clinical manifestations of FXS. I can only think about seizures that are present in childhood and tend to resolve over time. However, increase levels of sAPPα have been described in severe ASD, AD, and other neurodegenerative disorders which do not present seizures.
Since this article contributes to a new FXS therapeutic area, I feel that the authors should add much more information about the compounds used (TAT-Pro ADAM10 709-729) regarding in what models of diseases this compound has been used (Huntington's, AZ, ASD, etc), what phenotypes were rescued (animal behavior or cellular physiology), and to what extent this compound has been used in human trials if any. This will illustrate the overall sAPPα relevance and specificity to FXS. Similarly, the authors should state the limitations/precautions/benefits of the use of TAT-Pro ADAM10 709-729 in humans with FXS.
Author Response
Reviewer 1
Comments and Suggestions for Authors
The researcher demonstrated an age-dependent regulation of APP metabolism in different FXS cellular models (fibroblasts, iPSCs, and brain organoids). Particularly, sAPPα is involved in the APP-mediated increase of protein synthesis in FXS. This work identified the early stages of childhood and adolescence in humans as crucial time window for therapeutic intervention, based on the restoring of protein homeostasis acting on the regulation of sAPPα release. More importantly, they proved that a subset of FXS individuals could benefit from treatment with the cell-permeable peptide TAT-Pro ADAM10709-729 able to specifically reduce sAPPα release normalizing protein synthesis.
The experiments conducted were appropriate for their hypothesis and the figures illustrate very clean results, which are very consistent across the cellular models, not only in the qualitative areas but also related to the timing/developing of the tissues.
We thank the reviewer for the positive comments on our manuscript.
If there were any negative results in other experiments, this should be briefly included. We thank the reviewer for this suggestion. The previous manuscript had 2 “data not shown”. The first one (pg 8 revised manuscript) describing a significant increase of APP in FXS that we prefer to keep as data not shown because it is a confirmation and expansion (additional fibroblast cell lines) of a previous publication (Pasciuto et al., 2015). The second data not shown (pg 8 previous version) has been removed from the revised manuscript because this data refer to a correlation with only 5 patients. In addition, we have revised Figure 2 and included the absence of correlation between sAPPα and 3 Vineland scores (communication, socialization, and adaptive behaviour) for the entire cohort.
Please include a reason for not including morphology and electrophysiology data, given that this pathway was associated with hyperexcitability.
We thank the review for this comment. We are aware that 1) FXS iPSC-derived neurons display hyperactivity and deficits in action potential firing and spontaneous synaptic activity (reviewed in Lee et al., 2022); 2) the dysregulation of APP metabolism has been associated with neuronal hyperexcitability mainly when the amyloidogenic pathway is enhanced (Martinsson et al., 2022; Targa Dias Anastacio et al., 2022; Westmark et al., 2016a).
Here we focused on APP non-amyloidogenic pathway that we found altered only in neuronal precursor cells (NPCs), which are not mature cells with very low spontaneous activity. Since we did not observe significant alterations of sAPPα release in human mature neurons (day 60) we did not go further in the investigation of spine morphology and neuronal activity in this cell type. Since increased levels of Aβ have been detected in FXS murine model and FXS human brains (McLane et al., 2019; Pasciuto et al., 2015; Westmark et al., 2011, 2016b), it may be argued that hyperexcitability in mature FXS neurons may depend on a dysregulated amyloidogenic pathway.
The statistical analysis is lacking sample size calculations and the 2.9 Statistics section should be expanded to include p-value corrections and other details about the analysis.
We thank the reviewer for raising this issue. We added the sample size calculation to the Statistics section and revised the statistics (M&M pages 7-8), the Student’s t-test and the Pearson correlation analysis have been replaced by Mann-Whitney test and Spearman correlation analyses, respectively. Figure 3, Figure 4 and Supplementary Figure 1 and Figure legends have been modified accordingly.
The authors stated clearly the limitations of the study, I will appreciate a statement about the rigor and reproducibility of the research. There may be a need for another lab to replicate these results and this should be addressed/mentioned in the paper.
We thank the reviewer for this comment. We think it is indeed important to underline that the present work is based on a relatively small sample size for the clinical correlation analysis.
We based the present study on a heterogeneous group of 32 FXS individuals (four different cohorts) with different genetic backgrounds presenting mosaics or complete penetrance (see supplementary Table 1), derived from four different countries, the reproducibility should be warranted in other laboratories with genetically comparable cohorts. All the molecular analyses have been performed using standardized methods and for each cell line we analyzed the protein levels in technical replicates. In addition, some of the western blots have been performed by two independent co-authors. We are confident that, under the conditions described in our manuscript, the data are reproducible.
However, further studies on additional cohorts and performed by other laboratories are required for correlation studies in the age-stratified subgroups to envision a future clinical study. We have expanded in the discussion on the reproducibility and limitations of this study (pg 16.
While the authors try to link the levels of sAPPα with severity, it is hard to make a biological link between the age-related sAPPα defect and the clinical manifestations of FXS. I can only think about seizures that are present in childhood and tend to resolve over time. However, increase levels of sAPPα have been described in severe ASD, AD, and other neurodegenerative disorders which do not present seizures.
We thank the reviewer for this comment. Indeed, several FXS clinical phenotypes show considerable changes across ages in term of severity, such as attention problems, socialization defects, aggressive behavior and brain structural abnormalities (Cregenzán-Royo et al., 2022; Ellis et al., 2020; Quintin et al., 2016; Roberts et al., 2019). A possible biological link could be indeed the generation of seizures that are present in childhood and tend to resolve over time. Seizures have been reported in ASD (Buckley and Holmes, 2016; Kwon et al., 2022), Angelman Syndrome (Leung and Ring, 2013; Samanta, 2021) and AD (Born, 2015; Yang et al., 2022). Such a temporal excess of sAPPa could also have an impact on social behavior, that appears to improve during development (Cregenzán-Royo et al., 2022; Ellis et al., 2020; Roberts et al., 2019; Usher et al., 2020). Consistent with data generated in the mouse model of FXS, the temporal sAPPa upregulation has an impact on a social activity (nest building) that is rescued decreasing the excess of sAPPa (Pasciuto et al., 2015). We have now added a sentence in the discussion highlighting these possibilities (pg 16-17).
Since this article contributes to a new FXS therapeutic area, I feel that the authors should add much more information about the compounds used (TAT-Pro ADAM10 709-729) regarding in what models of diseases this compound has been used (Huntington's, AZ, ASD, etc), what phenotypes were rescued (animal behavior or cellular physiology), and to what extent this compound has been used in human trials if any. This will illustrate the overall sAPPα relevance and specificity to FXS. Similarly, the authors should state the limitations/precautions/benefits of the use of TAT-Pro ADAM10 709-729 in humans with FXS.
We thank the reviewer for this suggestion. We have now expanded this part in the discussion (pg 18-19) incorporating the use of peptide therapy in other models/diseases, its limitations and benefits.

Reviewer 2 Report
This manuscript, Cencelli et al., demonstrates age-dependent APP processing in fibroblast derived from FXS individuals, iPS-derived neural precursor cells and forebrain organoids. Authors also used cell permeable peptide to decrease the generation of sAPPa and showed the restoration of protein synthesis levels. Experiments are well-planned and well-written throughout the manuscript. Specifically, authors used extensive number of samples of FXS individuals with variety of culture models: fibroblasts derived from FXS individuals, iPS-derived neural precursors, as well as forebrain organoids. In addition, although there were previous studies demonstrating of APP, ADAM10 and sAPPa dysregulation in FXS fibroblasts, Cencelli et al., analyzed APP metabolism in a larger sample size with various age groups. Authors further demonstrated that with treating TAT-pro, it reduced ADAM10 localizes at cell surface, and the synaptic and behavioral deficits in Fmr1 KO mice can be rescued. Overall, manuscript is thoroughly well-written but there are still a few comments and questions to help improving the manuscript and for the clarifications.
Major comments:
- In Figure 2B, authors only presented group1 sAPPa levels/socialization score correlation; in the paragraph (pg 8, line 224-225), authors mentioned that there were no differences observed in groups 2 and 3 (data not shown). Can authors present groups 2 and 3 data (either next to Figure 2b or as a supplementary figure) instead ‘data not shown’? Also, please discuss about authors’ interpretation of why group1 might show the difference but groups 2 and 3 didn’t in the Result section or in the Discussion section. Especially in Figure 2A, authors plotted all groups in one graph, but not for Figure 2B. If there is any specific reason why authors wanted to split groups only for Figure 2B, please clarify in the Result section.
2. In Figure 3D, authors separated FXS to two groups, normal and high protein expression groups. Although it is acknowledged that there are groups of FXS fibroblast showing high protein expression and normal protein expression, there was no statement if this could be related to any age groups or not. Are these protein expression levels dependent on the age group or not related to age group at all? In other words, does age play a role in protein expression levels in FXS? It would be helpful if authors mention whether protein levels in FXS fibroblast correlates with age group or not for readers.
- In Figure 4B, authors monitored/evaluated sAPPa levels during neuronal differentiation. It is confusing and unclear what ‘Day’ status they are in Figure 4B. Please state clearly what ‘Day’ those samples are from for Figure 4b in Figure and Figure legends. For example, if they are matched with Figure 4A and are supposed to be Day 0, 6, 24 and 60, please clearly indicate that in the Figure itself and Figure legends for readers.
4. Authors treated TAT-pro to fibroblast but not tired on iPSC, NPC or neuron. (1) Do authors detect similar finding regarding total APP level increase in FXS in their 2D culture NPCs and neurons? (2) Can authors justify the reason why this (TAT-pro treatment) was exclusively done in fibroblast in their Result or Discussion section? Is this mainly due to the technical limit (delivery efficiency)? If so, what is the efficiency in fibroblast and neurons if they are known?
- Authors demonstrated a sAPPa increase in <30 yr fibroblast (Group 1 and 2) and also in NPCs (Day 24). Given that the dysregulation of non-amygdaloid pathway in FXS switch to amyloidogenic pathway after adult, do authors observe or predict that older age group or older NPCs can benefit more significantly from having TAT-pro treatment to prevent potential amyloid beta formation? Please discuss about this in the Result or Discussion section.
Minor comments:
There are a few errors throughout the manuscript. For example, (1) Pg 4, line 101: “TDI” should be ‘typically developing individual’ not ‘typically developing’. (2) authors need to be more consistent in notation of statistical significancy, e.g. in Figure 1 authors use “p<0.05 (no space between p and <)” but in Figure 3 “p < 0.001 (a space between p and <)”. (3) “coomassie blue” in Figure 4 legends needs to be “Coomassie blue”. Overall, authors should check spells and consistency in terms of nomenclature throughout the manuscript while revising.
Author Response
Reviewer 2
This manuscript, Cencelli et al., demonstrates age-dependent APP processing in fibroblast derived from FXS individuals, iPS-derived neural precursor cells and forebrain organoids. Authors also used cell permeable peptide to decrease the generation of sAPPa and showed the restoration of protein synthesis levels. Experiments are well-planned and well-written throughout the manuscript. Specifically, authors used extensive number of samples of FXS individuals with variety of culture models: fibroblasts derived from FXS individuals, iPS-derived neural precursors, as well as forebrain organoids. In addition, although there were previous studies demonstrating of APP, ADAM10 and sAPPa dysregulation in FXS fibroblasts, Cencelli et al., analyzed APP metabolism in a larger sample size with various age groups. Authors further demonstrated that with treating TAT-pro, it reduced ADAM10 localizes at cell surface, and the synaptic and behavioral deficits in Fmr1 KO mice can be rescued. Overall, manuscript is thoroughly well-written but there are still a few comments and questions to help improving the manuscript and for the clarifications.
We thank the reviewer for the positive comments on our manuscript.
Major comments:
- In Figure 2B, authors only presented group1 sAPPa levels/socialization score correlation; in the paragraph (pg 8, line 224-225), authors mentioned that there were no differences observed in groups 2 and 3 (data not shown). Can authors present groups 2 and 3 data (either next to Figure 2b or as a supplementary figure) instead ‘data not shown’? Also, please discuss about authors’ interpretation of why group1 might show the difference but groups 2 and 3 didn’t in the Result section or in the Discussion section. Especially in Figure 2A, authors plotted all groups in one graph, but not for Figure 2B. If there is any specific reason why authors wanted to split groups only for Figure 2B, please clarify in the Result section.
We thank the reviewer for this suggestion, we have revised Figure 2 removing the age-subgroup correlation analysis, because of the small sample size (n = 5-8) of the groups. Revised Figure 2 shows the correlation between sAPPα levels and the 4 VABS scores in the FXS cohort for which the clinical data were available (n = 18 subjects). Results and discussion have been modified accordingly (see pg 8-9 and pg 18-19). Below the revised Figure 2.
- In Figure 3D, authors separated FXS to two groups, normal and high protein expression groups. Although it is acknowledged that there are groups of FXS fibroblast showing high protein expression and normal protein expression, there was no statement if this could be related to any age groups or not. Are these protein expression levels dependent on the age group or not related to age group at all? In other words, does age play a role in protein expression levels in FXS? It would be helpful if authors mention whether protein levels in FXS fibroblast correlates with age group or not for readers.
We thank the reviewer for raising this point. The variability observed in the levels of protein synthesis in FXS individuals was previously shown to be age-independent in fibroblasts (Jacquemont et al., 2018; Kumari et al., 2014).
Following the reviewer’s comment, we investigated if such a correlation could be detected in the cohort analyzed in this present study, although it consists mostly of subjects enrolled in Jacquemont et al, 2018 with the addition of a few cell lines. As shown in Figure 1 for the reviewer only, no correlation was detected. We have modified the text accordingly in the Results section (pg 12 and 15). Below Figure 1 for the reviewer only.
- In Figure 4B, authors monitored/evaluated sAPPa levels during neuronal differentiation. It is confusing and unclear what ‘Day’ status they are in Figure 4B. Please state clearly what ‘Day’ those samples are from for Figure 4b in Figure and Figure legends. For example, if they are matched with Figure 4A and are supposed to be Day 0, 6, 24 and 60, please clearly indicate that in the Figure itself and Figure legends for readers.
We apologize for the lack of clarity. The revised Figure 4 now includes the information of the “Day” in panel B, the legend is also modified accordingly.
- Authors treated TAT-pro to fibroblast but not tired on iPSC, NPC or neuron.
(1) Do authors detect similar finding regarding total APP level increase in FXS in their 2D culture NPCs and neurons?
We do not detect a significant difference in total APP in NPCs (day 24) while we do at day 60 (Figure 2 for the reviewer only) suggesting that a switch in the dysregulation of the amyloidogenic pathway at later stages. While the study of the switch to the amyloidogenic pathway during FXS neuronal differentiation is interesting and worth investigating, we believe it is not the scope of this current work but a question to address with future investigations.
(2) Can authors justify the reason why this (TAT-pro treatment) was exclusively done in fibroblast in their Result or Discussion section? Is this mainly due to the technical limit (delivery efficiency)? If so, what is the efficiency in fibroblast and neurons if they are known?
We thank the reviewer for this comment. The choice to use fibroblasts as a model to test the TAT-pro peptide responds to a precise study design. Due to the heterogeneity of the FXS clinical and molecular phenotype, to design new clinical trials it is crucial to stratify patients into subgroups aiming at personalized medicine for this syndrome. In this respect, compared to iPSC-derived NPCs and neurons, fibroblasts meet well this criterion while the reprogramming returns adult cells to the stem cell state that is independent from the initial age of the individuals (Ernst, 2020). We have added a comment on the use of fibroblasts and NPCs/neurons in the Results (pg 15) and in the Discussion (pg 22) sections.
Following the reviewer’s comment, we performed a preliminary analysis that revealed that the TAT-Pro peptide treatment reduces sAPPα release in FXS NPCs (Figure 3 for the reviewer only). Further investigations are required to confirm this data set, importantly we believe that the use of fibroblasts, that allowed us to identify an age-specific subgroup of FXS individuals with dysregulated sAPPα, remains the optimal cell type for patient stratification and “proof of concept” use of the TAT-Pro peptide. Below the Figure 3 for the reviewer only.
- Authors demonstrated a sAPPa increase in <30 yr fibroblast (Group 1 and 2) and also in NPCs (Day 24). Given that the dysregulation of non-amygdaloid pathway in FXS switch to amyloidogenic pathway after adult, do authors observe or predict that older age group or older NPCs can benefit more significantly from having TAT-pro treatment to prevent potential amyloid beta formation? Please discuss about this in the Result or Discussion section.
We thank the reviewer for this comment. Favoring the switch to the non-amyloidogenic pathway is an attractive therapeutic strategy that has been explored in animal models of Alzheimer’s Disease (AD) (Habib et al., 2017; Mockett and Ryan, 2022). For example, the subchronic treatment of WT mice with the TAT-Pro peptide has been used to generate a model of sporadic AD favoring the release of Aβ peptide and formation of β-amyloid aggregates (Epis et al., 2010) (see discussion at pg 23). We therefore predict that older patients will not benefit more from the TAT-Pro peptide treatment because at an old age the ADAM10 dysregulation is not present anylonger and such a treatment would boost the production of Aβ peptides.
Therefore, a personalized therapy for FXS should rely on an age-dependent stratification of FXS subjects based on sAPPα and Aβ levels. The TAT-Pro peptide should be given during young age/adolescence before the APP metabolism switches generating an excess of Aβ at an older age (McLane et al., 2019; Westmark et al., 2016).
Minor comments:
There are a few errors throughout the manuscript. For example, (1) Pg 4, line 101: “TDI” should be ‘typically developing individual’ not ‘typically developing’. (2) authors need to be more consistent in notation of statistical significancy, e.g. in Figure 1 authors use “p<0.05 (no space between p and <)” but in Figure 3 “p < 0.001 (a space between p and <)”. (3) “coomassie blue” in Figure 4 legends needs to be “Coomassie blue”. Overall, authors should check spells and consistency in terms of nomenclature throughout the manuscript while revising.
We thank the reviewer for these suggestions, the manuscript has been carefully revised for errors and nomenclature consistency.

Round 2
Reviewer 1 Report
Thank you much for addressing all the comments, the responses are thorough and appropriate. I appreciated the references included.